# Fiscal decentralization, local government innovation preference, and enterprise technological innovation: Evidence from China

Jie Tu [1,2,☯,¤a,¤b,*]

1 School of Digital Sport, Quanzhou College of Technology, Quanzhou, Fujian Province, China, 2 School of Economics and Finance, Huaqiao University, Quanzhou, Fujian Province, China

☯ This authors contributed equally to this work.
¤a Current Address: School of Digital Sport, Quanzhou College of Technology, Quanzhou, Fujian, China
¤b Current Address: School of Economics and Finance, Huaqiao University, Quanzhou, Fujian, China
* 13023924134@163.com

**Citation:** Tu J (2026) Fiscal decentralization, local government innovation preference, and enterprise technological innovation: Evidence from China. PLoS One 21(4): e0347025. https://doi.org/10.1371/journal.pone.0347025

## Abstract

Fiscal decentralization influences the allocation of attention and effort by local governments across multiple tasks, consequently impacting enterprise technological innovation within their jurisdictions. This represents a significant institutional constraint on corporate innovation. Employing a sample of A-share listed companies that regularly published annual financial reports from 2015 to 2023, this study constructs a moderated mediation model within a three-tier "central government – local government – enterprise" principal-agent framework. It is the first to uncover the intrinsic mechanism and property-right heterogeneity through which fiscal decentralization affects the technological innovation of microeconomic entities. The empirical findings reveal: (1) Overall, fiscal decentralization exerts a significant inhibitory effect on enterprise technological innovation, providing micro-level empirical support for the "decentralization inhibition theory". (2) Local government preference for innovation serves as a mediator between fiscal decentralization and enterprise technological innovation. Fiscal decentralization indirectly influences enterprise innovation by shaping local governments' innovation priorities. (3) The mediating effect of local government innovation preference exhibits property-right dependence. Fiscal expenditures on science and technology and innovation incentive policies by local governments trigger arbitrage behavior in private enterprises, creating a "crowding-out effect" on their technological innovation. As "special market entities," state-owned enterprises (SOEs) combine the advantages of both market and governmental economic coordination mechanisms. Fiscal decentralization, by influencing local government innovation preference, effectively promotes technological innovation in SOEs. These empirical results offer policy implications for accelerating the formation of new institutional frameworks compatible with new quality productive forces, particularly concerning central-local relations, government-enterprise relations, and the functional positioning of SOEs.

**Data availability statement:** All relevant data are within the paper and its Supporting Information files.

**Funding:** The author(s) received no specific funding for this work.

**Competing interests:** NO authors have competing interests.

## 1. Introduction

Under the background of diminishing marginal effect of monetary policy, the importance of finance for economic development and scientific and technological innovation has become the consensus of governments. As a national fundamental institutional arrangement of fiscal power distribution relationship [1], the in fiscal decentralization China implies a three-layer principal-agent relationship among the central government, local governments and enterprises [2,3], which constitutes a de facto institutional constraint on the technological innovations of the enterprises through the government's "tangible hand". Second, due to differences in political and economic environments, fiscal decentralization practices vary across countries. In China, fiscal decentralization has played a particularly important role [4]. The coexistence of political centralization and economic decentralization ensures central government control while aligning local government incentives with market economic development [5,6]. Decentralization reforms require sufficient institutional and state capacity [4]. In Russia, the central government has gradually lost leadership during decentralization and failed to provide sufficient incentives for local development [7]. In many developing countries, decentralization is associated with insufficient public resource supply and increased inequalit. Against this backdrop, examining China's local and county fiscal system provides practical experience for other centrally planned countries facing decentralization challenges.

This paper analyzes the existing literature from the technical innovation effect of fiscal decentralization and the influence factors of enterprise technological innovation in two aspects: First, the technical innovation effect of fiscal decentralization. The effect of fiscal decentralization on technological innovation is mainly analyzed by provincial and municipal local governments [8], and there are two distinct views: "decentralization inhibition theory" and "decentralization promotion theory" [9]. Scholars who support the "decentralization promotion theory" believe that fiscal decentralization is conducive to strengthening the government's innovation function [10], and promoting the development of technological innovation activities of local governments and their efficiency [11]. Scholars who support the "decentralization inhibition theory" believe that fiscal decentralization will exacerbate the vicious competition between governments and make the investment structure of local governments show the obvious self-interested characteristic of "light on innovation and heavy on production" [9,12,13]. This inhibitory effect does not stem from decentralization itself, but rather from the promotion tournament mechanism embedded in China's hierarchical governance system. When local officials are constrained by limited time, energy, and resources, and their career advancement depends heavily on relative performance evaluation set by higher authorities—especially GDP growth—they tend to prioritize short-term, quantifiable productive investments over innovation inputs characterized by high uncertainty and long cycles [14,15]. However, when the central government adjusts the evaluation criteria and strengthens innovation indicators, the same fiscal decentralization system can instead incentivize local governments to engage in an innovation tournament [1]. The reason why fiscal decentralization reduces the local government's innovation preference is that, on the one hand, political competition

among Chinese local officials is structured as a performance-based tournament overseen by higher-level authorities. Yu et al. provide systematic evidence that such competition leads to spatial interactions among politically connected jurisdictions, as local officials strategically adjust their behavior in response to their peers in order to improve their relative standing in the promotion hierarchy [14,15]. Compared with the productive financial expenditures such as infrastructure, the innovative expenditures have the attribute of public goods, and this kind of behavior of "doing others a favor" is not in line with the interests of the local government. On the other hand, scientific and technological innovation is highly uncertain. Under the promotion championship mechanism of political centralization and economic decentralization, local government officials with the attribute of "politicians" tend to squeeze out the expenditures on unproductive public goods such as innovation in pursuit of economic growth during their tenure [12].

Secondly, the influencing factors of corporate technological innovation. Research on the factors influencing corporate technological innovation takes corporate governance as the main perspective. Yermack and Zahra et al. found that the relationship between the board size and corporate technological innovation shows an inverted "U" curve, which is due to the following reasons The reason is the "double-edged sword" effect of board size expansion [16,17]. On the one hand, the expansion of the board of directors is conducive to the complementarity of various professional knowledge and different opinions required for technological innovation, and reduces the risk of technological innovation; on the other hand, the expansion of the board of directors will lead to the increase of the difficulty of communication and coordination among the members of the board of directors, the phenomenon of board of directors' "free-riding", as well as the emergence of slow board of directors' decision-making, cumbersome procedures and other problems. On the other hand, an increase in board size may lead to increased communication and coordination among board members, "free-riding" by the board, and slow and cumbersome board decision-making. The optimal board size to promote technological innovation depends on the balance of the above positive and negative effects. Han and Jiang and Liu et al. studied the impact of shareholding structure on corporate technological innovation, and concluded that higher shareholding concentration inhibits technological innovation [18,19]. However, some scholars have come to the opposite conclusion [20], and the probable reason for the different conclusions of the study lies in the differences in the degree of shareholding checks and balances. Under the same high level of equity concentration, the low level of equity checks and balances places more emphasis on the sustained profitability of the firm and promotes technological innovation [21].

The above literature provides ideas inspiration and paradigm borrowing for the research of this paper, but there is still room for further deepening and expanding. This paper takes A-share listed companies that regularly publish annual financial reports between 2015 and 2023 as samples, and reveals for the first time the role of fiscal decentralization in technological innovation of microeconomic agents in the three-tier principal-agent framework of "central government-local government-enterprises". In the three-tier principal-agent framework of "central government-local government-enterprises", this paper reveals for the first time the intrinsic mechanism of fiscal decentralization and the heterogeneity of property rights in the technological innovation of microeconomic agents. It responds to the broader discussion on decentralization and innovation under different institutional contexts, and provides China's experience for countries with strong vertical accountability mechanisms.

Compared with the existing literature, the marginal contributions of this paper are:

(1) In terms of research perspective, it enriches and expands the academic scenario of the "central government-local government-enterprise" three-layer principal-agent model. The research on the three-tier principal-agent model of "central government-local government-enterprise" in the Chinese context is the first of its kind. The research on the three-tier principal-agent model was initiated by Nie and Zhang [22], who portrayed the equilibrium and preventive collusion contracts of the central government in the classic government-enterprise collusion phenomena such as mining accidents, environmental pollution, etc., and pointed out that the central government tends to acquiesce to the collusion of the local government and the enterprises under the decentralization (territorial management) system,

compared to the centralized (vertical management) system. On the basis of analyzing and demonstrating the relationship between the central government and local governments, this paper clarifies for the first time the effect and mechanism of fiscal decentralization on technological innovation under the perspective of new quality productivity, and enriches and expands the "central government-local government-enterprise" model, which is a three-tier principal-agent model in the context of China's major economic issues. It enriches and expands the application of the "central government-local government-enterprise" three-layer principal-agent model in China's major real-life economic problems, which can be effectively applied to other research areas where policy outcomes are determined by tensions in central–local relations. This study identifies the boundary conditions under which fiscal decentralization affects innovation, thereby providing a more refined theoretical perspective for understanding how institutional design shapes policy implementation across national contexts.

(2) In terms of research methodology, on the basis of portraying the degree of fiscal decentralization by the index of fiscal autonomy, the instrumental variables of fiscal decentralization are constructed by the cross-multiplication of geographic distance and time trend from provincial capitals to Beijing (the central government), so as to make the identification of the causative relationship between fiscal decentralization and technological innovation of enterprises cleaner and more credible; furthermore, a two-step regression model is set up with the regulated mediator model by adopting the two-step regression method proposed by Zhao et al. [23]. method to set up a mediation model with regulation to test the mechanism of fiscal decentralization on the technological innovation of enterprises and the moderating effect of the nature of property rights, so as to provide relevant research with empirical methods and empirical evidence that can be used as reference.

(3) In terms of research content, the nature of property rights is introduced into the study of enterprise technological innovation, to clarify the major misunderstandings such as "the attribute of public property rights of state-owned enterprises determines their loss of innovation efficiency". As a "special market subject" with the dual attributes of profit-making and fulfillment of national mission, it is clear that SOEs play an exemplary role in technological innovation and a key leading role in the formation and development of new quality productive forces.

## 2. Theoretical analysis and research assumptions

From the perspective of the evolution path of China's fiscal system, before the 1994 tax-sharing reform, the fiscal system mainly acts directly on microeconomic subjects through embedding enterprises (especially state-owned enterprises). The tax-sharing reform has embedded China's finances in the economic cycle, forming a three-tier principal-agent relationship among the central government, local governments and enterprises. Local governments play a pivotal role in connecting the central government and enterprises in their jurisdictions. While the three-tier principal–agent framework provides a useful analytical lens, its implications depend critically on the institutional context in which decentralization operates [4]. The local governance literature emphasizes that the effects of fiscal decentralization are mediated by vertical accountability mechanisms, local administrative capacity, and the degree of policy discretion granted to subnational governments [1]. Therefore, based on the current system, there is an obvious logical break in the study of the direct impact of fiscal decentralization on technological innovation of enterprises. This paper focuses on the mechanism of Chinese fiscal decentralization on technological innovation of enterprises and the characteristics of property rights dependence through the construction of a moderated intermediation model.

### 2.1. Fiscal decentralization and technological innovation of enterprises

As an important institutional arrangement to adjust the allocation of financial resources between the central government and the local government, fiscal decentralization guides enterprises in technological innovation by enhancing the local

government's support for science and technology [10]. Fiscal decentralization endows local governments with fiscal autonomy and information advantages, which in theory can support the implementation of locally tailored innovation policies [10,11]. However, under the dual context of political centralization and economic decentralization in China [5,6], local officials face a multi-task principal-agent problem, and promotion incentives dominate resource allocation decisions [24]. When the performance evaluation system is heavily biased toward GDP growth, fiscal autonomy enables local governments to shift resources to infrastructure and production capacity construction that yield immediate economic returns, thereby crowding out innovation inputs. In fact, this is not contradictory to the logic that local governments carry out innovative activities to obtain "points" from the state, further indicating that the definition of such "point-scoring" behavior depends on the assessment criteria set by the central government.

However, due to the high uncertainty of technological innovation, it is worth pondering whether fiscal decentralization necessarily promotes technological innovation of enterprises. Enterprises may face the following problems in the process of carrying out technological innovation activities: firstly, technological innovation is a trial-and-error process, and the marginal productivity of R&D investment in the pre-innovation period is low, so it is difficult for enterprises to obtain substantial output returns, which will generate negative feedback on the subsequent development of innovation activities [25]. Second, technological innovation is the exploration of the feasibility of new technologies. In order to ensure the profitability of technological innovation and set up innovation barriers to prevent potential competitors from entering, enterprises need to continuously track the technological frontier, but they may fall into the capability trap of "over-exploration" [26]. Finally, in an era characterized by exponential technological advances, firms may have the problem of underestimating the rate of technological obsolescence. In the era of knowledge economy, the speed of technological update is too fast, and it is often difficult for enterprises to accurately judge when to change or stop their investment strategy in technological innovation, and thus they are more inclined to increase their R&D investment in existing technologies, which is not conducive to the output of high-quality technological innovation results. The knowledge limitations of local governments in the supply of technological innovation policies are also an important inhibiting factor for enterprise technological innovation under the current fiscal decentralization system. Furthermore, in settings characterized by strong vertical accountability but limited local administrative capacity, decentralization may incentivize local governments to prioritize short-term, measurable outcomes over long-term innovation investments, which are inherently uncertain and difficult to evaluate [1,15].

Wang et al. and Xie and Wu found that enterprises in the growth and decline stages focus on "survival", while enterprises in the maturity stage focus on "development" [27,28]. Mature enterprises tend to invest in innovative projects with high capital investment and long payback period but high expected returns [29]. Therefore, financial subsidies and tax incentives mainly have an incentive effect on the technological innovation behavior of enterprises in the mature stage of the life cycle, and have a smaller incentive effect on the technological innovation of enterprises in the growth and decline stages. In addition to the time dimension (enterprise life cycle stage), there is also significant heterogeneity in the innovation policy incentive effects in the cross-sectional dimensions of industry attributes, enterprise size, and the degree of market competition [24,30,31]. However, local governments often implement "one-size-fits-all" policy design, ignoring the differences and autonomous selectivity of enterprises, and even directly "circle the winners" with selective and non-universal policies, thus failing to form effective incentives for technological innovation. Comprehensive analysis of the above dual perspective of the enterprise and the government, put forward the following hypothesis:

Hypothesis H1: fiscal decentralization has a significant inhibitory effect on enterprise technological innovation.

## 2.2. The mechanism of fiscal decentralization on enterprise technological innovation and its heterogeneity

### 2.2.1. Fiscal decentralization and local government innovation preference.
China's first economic decentralization was launched in 1958, and although it did not achieve the expected results due to the impact of the Great Leap Forward, the large-scale multi-dimensional decentralization objectively constituted a kind of institutional experiment

[10], providing a blueprint for subsequent reforms in a path-dependent manner. 1994 saw the nationwide implementation of the tax-sharing reform, which marked the beginning of the fiscal decentralization based on standardized tax divisions. the formation of a fiscal decentralization system based on a standardized tax division, and local governments have since gained relatively stable and independent economic autonomy. Unlike economic decentralization, according to the formal institutional expression of the constitution, China is still a typical centralized unitary state. Its main features are: local government power is delegated by the central government; provincial and non-provincial key officials are appointed by the central government, and other local officials are appointed from the top down; the hierarchical power matrix between the different levels of government, and vertical inter-governmental interactions follow a "command-and-obey" pattern of behavior. In China's hierarchical governance system, local officials are embedded in a promotion tournament characterized by relative performance evaluation rather than electoral accountability [15]. Promotion prospects depend critically on how well local leaders perform relative to their political peers, especially in terms of indicators emphasized by the central government. As shown by Yu et al., this political competition generates strong strategic interactions among local governments, particularly among politically proximate jurisdictions, and induces local officials to align their policy actions with central priorities in order to accumulate "points" in the promotion tournament [14].

Therefore, the process of economic decentralization in China is also the process of local governments accepting the central government's performance appraisal and carrying out "political championships" [32]. The government is a typical multitasking political organization, in which the local government, as an agent, needs to trade-off between multiple potentially conflicting tasks, while the central government, as a principal, needs to guide the local government to rationally allocate its attention and effort among multiple tasks through the design of an optimal incentive contract [33]. For a long time, the central government has mainly used simple and measurable relative GDP growth indicators to assess the performance of local officials, resulting in a distorted allocation of local governments' efforts among multiple tasks (Hall and Maffioli, 2008), and creating a "crowding out effect" on technological innovation. "the proportion of the added value of core industries in the digital economy to GDP" and other technological innovation indicators have been successively incorporated into the government's work objectives, while the GDP growth rate indicator continues to be downwardly adjusted or even blurred (qualitative expression). By changing the relative incentive strengths of economic growth and technological innovation, the central government has corrected the distortion of local governments' attention and efforts in the multitasking political organization, and guided local governments to shift from "competing for growth" to "competing for innovation" [1]. In this study, local government innovation preference is conceptualized as an institutionalized policy orientation revealed through fiscal allocation decisions, rather than a psychological inclination of individual policymakers.

**2.2.2. Local government innovation preference and enterprise technological innovation and property rights heterogeneity.** Technological innovation has the attribute of public goods, and the private benefits that innovation subjects can obtain from it are lower than the social benefits, resulting in the output and efficiency of enterprise innovation not reaching the optimal level. In order to overcome the "market failure" caused by the positive externalities of technological innovation, the government may need to increase its fiscal expenditure on science and technology and make concerted use of policy tools such as financial subsidies and tax incentives to motivate enterprises to strengthen technological innovation. Because of the complexity of the behavioral responses of enterprises in economies in transition, local government innovation incentives may promote substantive technological innovation or become a strategic tool for enterprises to extract policy resources.

The existing literature generally follows the analytical framework of neoclassical economics and treats State-owned enterprises as general market players, arguing that the expansion of State-owned property rights is inefficient in areas that require sustained innovation [34]. In practice, however, SOEs have been an important tool in many countries around the world to address "market failures" caused by private returns falling short of social returns. Emerging economies in Latin America, such as Argentina, Brazil, Chile, Colombia, Mexico, etc and developed economies, such as France, Norway, Austria, etc. have used SOEs as a way to channel the development of basic research or high-risk emerging industries

to cope with the shortage of innovation supply [35,36]. In China, SOEs have also replaced state-led research institutes and higher education institutions as the new "technology diffusion centers". The fundamental reason why SOEs play an important role in technological innovation is that, as the core institutional arrangement of the socialist market economic system [37], SOEs, in addition to the pursuit of profits in the sense of financial accounting, the national rent, i.e., the national mission, is also an important goal of their behavior. At the same time, extensive evidence suggests that SOEs are often associated with inefficiencies, soft budget constraints, rent-seeking behavior, and governance challenges, which may undermine innovation efficiency if not properly addressed [37,38]. Although the non-market contractual characteristics of SOEs are a great dissolution of the error correction function of the market mechanism, through a series of sustained market-oriented reforms such as the mixed ownership reform and the reform of dividend rights incentives, SOEs can combine the advantages of the market and the government's two modes of economic coordination to partially mitigate certain forms of market failure and government failure under specific institutional and governance conditions. However, such institutional arrangements cannot eliminate government failure, nor can they avoid the efficiency losses associated with the nature of state-owned property rights. As an institutional trade-off, the innovation performance of SOEs is closely tied to the quality of governance and the depth of reforms [37].

Unlike SOEs, a significant portion of private enterprises are not motivated by the local government's preference for innovation, but rather by access to government financial resources to ease financing constraints and operational pressures. An important consequence is that the policy arbitrage behavior of private enterprises forms a de facto "political asylum", which in turn interferes with their technological innovation decisions. According to the theory of strategic blocking, as long as there are long-term profits, incumbent firms may be able to block the entry of potential competitors through technological innovation. However, the "political patronage" provided by the government in terms of finance (S&T spending, government subsidies, tax incentives), land, credit, market access (investment restrictions, project approval rights, business licensing rights), etc. can hedge against the competitive pressure in the market and reduce the threat of entry of incumbent firms, so that the policy arbitrage becomes a "viable" alternative for the incumbent firms' technological innovation deterrence, which is a "viable" alternative for the incumbent firms. A "viable" alternative [39,40]. According to the logic of collective action, in an innovation race between multiple incumbents and potential entrants, each incumbent wants to enjoy the benefits of the others without paying the costs (i.e., "free-riding"), resulting in the higher the barriers to entry, the lower the incentives for firms to technologically innovate [41].

Accordingly, this paper argues that the role of SOEs in the field of innovation is conditional and context-dependent, rather than an indication of their inherent superiority over private enterprises. Integrating the analysis of the mechanism of fiscal decentralization on the role of enterprise technological innovation and property rights heterogeneity, the following hypotheses are proposed:

Hypothesis H2: Local government innovation preference has a mediating role between fiscal decentralization and enterprise technological innovation, and fiscal decentralization indirectly affects enterprise technological innovation by influencing local government innovation preference.

Hypothesis H3: The mediating role of local government innovation preference is characterized by property rights dependence, and fiscal decentralization promotes (inhibits) technological innovation of state-owned (private) enterprises through influencing local government innovation preference.

## 3. Research design

### 3.1. Variable selection and measurement

**3.1.1. Explained variables.** The explanatory variable in this paper is enterprise technological innovation (*Innovation*). Technological innovation is the core driving force of new quality productivity. Part of the literature adopts technological innovation input to measure the level of technological innovation of enterprises [42]. However, sufficient technological

innovation input is only a prerequisite and a means for enterprises to carry out innovative activities, and high-quality technological innovation output is the ultimate goal. Compared with general investment activities, enterprise technological innovation is characterized by long cycle and high risk, and whether enterprises can transform technological innovation inputs into high-quality technological innovation outputs has great uncertainty, so technological innovation inputs can not well characterize the level of enterprise technological innovation. Patent is an important indicator reflecting the technological innovation output of enterprises, which is characterized by easy accessibility, comparability and large amount of information. This paper draws on the practice of Wu and Wu and adopts the number of patent applications to measure the technological innovation of enterprises [43]. This paper defines technological innovation as the innovative outcomes achieved through the development and application of new technologies. Although patents may cover various fields, this study specifically focuses on patents related to new technologies, which directly contribute to a company's technological advancement and market competitiveness.

### 3.1.2. Explanatory variables.

The explanatory variable in this paper is fiscal decentralization (*Fisdec*). The existing literature mainly uses four types of indicators to measure fiscal decentralization:revenue and expenditure indicator [12], fiscal autonomy indicator/fiscal self-sufficiency rate [44], incremental own revenues [45], and composite indicator [10]. An appropriate decentralization indicator needs to reflect both the inter-temporal changes in the central-local relationship and the inter-regional differences, and there is no optimal measurement method yet. In this paper, we use the fiscal autonomy indicator to portray the degree of fiscal decentralization in China from 2015 to 2023 because (1) the fiscal autonomy indicator is better than the revenue and expenditure indicator and the increment of own revenue during the study period. Fiscal autonomy, revenue and expenditure, and incremental own revenues cannot cover the entire period after the founding of New China, but the fiscal autonomy indicator is applicable to the study after the 1994 tax reform and can reflect regional differences among provinces; (2) the fiscal autonomy indicator is better than the composite indicator within the study period. Scholars tend to use a variety of measures, but different weighting indicators reflect very different facts and logic behind them, and imply completely different policy implications. For example, Zheng et al. use both revenue decentralization indicators and expenditure decentralization indicators, and find that they have opposite effects on local government S&T investment [46]. Revenue decentralization significantly inhibits local government S&T investment, while expenditure decentralization has a significant promotion effect on local government S&T investment. Therefore, various types of indicators cannot be mixed [46], and there are some problems in combining them. Lv et al. believe that the core of Chinese fiscal decentralization is fiscal revenue decentralization [47], and measure the proportion of fiscal revenue distribution among the central, provincial, municipal, and county levels, covering tax revenue, non-tax revenue, and general public budget revenue of various revenue calibers, and take it as a measurement index of fiscal decentralization. Due to the limitation of statistical data sources, the time span of fiscal decentralization data calculated directly or indirectly using this method is only from 1994 to 2014. Therefore, following the research of Jiang et al., this paper still uses fiscal autonomy, measured as the ratio of local fiscal revenue to local fiscal expenditure, as an alternative indicator of China's fiscal decentralization [48]. Fiscal autonomy reflects the degree of discretion of local governments in formulating relevant fiscal policies. It affects the economy through local government preferences, expenditure constraints, and government bargaining power.

### 3.1.3. Mediating variable.

The mediating variable in this paper is local government innovation preference (Prefer), which is measured by the ratio of local fiscal science and technology expenditures to local fiscal general budget expenditures, with reference to the practice of Ye and Zeng [49]. The primary approaches through which governments support innovation activities include establishing institutional mechanisms to protect innovation, developing hardware and software environments, and providing direct fiscal support. The proportion of science and technology expenditure in total government fiscal expenditure reflects the degree of governmental participation in regional innovation initiatives. It represents tangible policy commitments and resource allocation decisions of the government, and exerts a more practical impact on enterprises' innovation activities.

**3.1.4. Moderating variables.** The regulating variable of this paper is the nature of property rights (State), referring to the practice of Huang et al., according to the results of the calculation of the chain of control of the shareholding, state-owned enterprises are assigned the value of "1", and private enterprises are assigned the value of "0" [50].

**3.1.5. Control variables.** The control variables in this paper are selected from the enterprise level and the regional level respectively. The control variables at the enterprise level are: equity concentration (Herf), measured by the sum of squares of the shareholding ratios of the top five shareholders; cash holdings (Cash), measured by the ratio of money funds to total assets; company size (Size), measured by the natural logarithm of total assets; debt-to-asset ratio (Debt), measured by the ratio of liabilities to total assets; and the age of company listing (Age), measured by the ratio between the year of attribution of the panel data and the age of private companies., measured using the natural logarithm of the difference between the year of attribution and the year of listing of the panel data. Regional level control variables: regional economic development level (GDP), measured by GDP per capital; openness level (FDI), measured by FDI per capital; degree of financial support (Finance), measured by the ratio of value added of the financial sector to GDP; and industry development level (Industry), measured by the ratio of value-added of the secondary sector in GDP. The names and descriptions of the variables are shown in Table 1.

## 3.2. Econometric model setting

In order to test the impact of fiscal decentralization on enterprise technological innovation, this paper sets the following econometric model:

$$Innovation_{i,t} = \alpha_0 + \alpha_1 Fisdec_{i,t} + \alpha_2 Herf_{i,t} + \alpha_3 Cash_{i,t} + \alpha_4 Size_{i,t} + \alpha_5 Debt_{i,t} + \alpha_6 Age_{i,t} + \alpha_7 GDP_{i,t}$$
$$+ \alpha_8 FDI_{i,t} + \alpha_9 Finance_{i,t} + \alpha_{10} Industry_{i,t}$$

(1)

Where i stands for individual company, t denotes annual identification; Innovation denotes enterprise technological innovation, Fisdec denotes fiscal decentralization; and the rest are control variables. In order to exclude the interference of extreme values on the regression results, this paper shrinks the upper and lower 1% of all continuous variables.

**Table 1. Variables and definitions table.**

| Variable type | Variables | Name | Description |
|---|---|---|---|
| Explained Variable | *Innovation* | Enterprise technological innovation | The number of patent applications |
| Explanatory Variable | *Fisdec* | Fiscal decentralization | The fiscal autonomy indicator (Local fiscal revenue/ local fiscal expenditure) |
| Mediating Variable | *Prefer* | Local government innovation preference | Local fiscal science and technology expenditures/ local fiscal general budget expenditures |
| Moderating Variable | *State* | Nature of property rights | 1 represents state-owned enterprises, 0 represents private enterprises. |
| Control Variable | *Herf* | Equity concentration | The sum of squares of the shareholding ratios of the top five shareholders |
| | *Cash* | Cash holdings | Money funds/ total assets |
| | *Size* | Company size | The natural logarithm of total assets |
| | *Debt* | Debt-to-asset ratio | Liabilities/ total assets |
| | *Age* | Age of company listing | ln (the year of attribution-the age of private companies) |
| | *GDP* | Regional economic development level | GDP per capital |
| | *FDI* | Openness level | FDI per capital |
| | *Finance* | Degree of financial support | Value added of the financial sector/ GDP |
| | *Industry* | Industry development level | Value-added of the secondary sector/ GDP |

### 3.3. Research sample and data source

This paper uses A-share listed companies that were listed prior to the end of 2016 and regularly issued annual financial reports between 2015 and 2023 as the initial sample, and progressively screens them according to the following criteria:

(1) Excluding companies listed after December 31, 2015 and delisted during the study period;

(2) Excluding financial companies such as money and financial services, capital market services, insurance, and other financial industries in the China Securities Regulatory Commission's (CSRC) 2021 version of industry classification;

(3) Excluding ST, *ST and other companies with abnormal financial status;

(4) Excluding companies whose ownership nature of the actual controller is not state-owned and private (e.g., collectively owned enterprises, Hong Kong, Macao and Taiwan-funded enterprises, foreign enterprises, institutions, social organizations, natural persons from Hong Kong, Macao and Taiwan, and natural persons from foreign countries, etc.) and companies whose ownership nature has changed during the study period;

(5) Excluding companies with missing data in the company/year for the research variables such as explanatory variables, explanatory variables, mediating variables, moderating variables, and control variables.

After the above steps, the study finally selected a sample of 9,180 companies/years. The financial data of the research sample are from the sub-databases of "R&D Innovation of Listed Companies", "Financial Statements" and "Analysis of Financial Indicators" in the CSMAR database, and the statistical data are from the statistics published by the National Bureau of Statistics (NBS). The statistical data come from "National Economic Accounting", "Population", "Foreign Economy and Trade", "Finance" and other statistical indicators published by National Bureau of Statistics (NBS). Descriptive statistics of main variables.

### 3.4. Descriptive statistics of main variables

The descriptive statistics of the main variables in this section are shown in the table below. From the Table 2, it can be seen that among the 9,180 companies/year samples, the mean value of technological innovation of enterprises (measured by the number of patent applications) stands at 107.918, with a median value of 82. This indicates a distinct right-skewed distribution of the sample, where more than half of the firms exhibit innovation levels below the sample average. The standard deviation of corporate technological innovation is 203.622, with a minimum value of 1 and a maximum value of 4,502. These figures further reflect the significant heterogeneity and high degree of dispersion in technological innovation outputs across firms. Following this logic, further found that more than 50% of Chinese provinces (including municipalities directly under the central government) have higher-than-average degrees of fiscal decentralization and local government innovation preferences, but there are large differences in the degree of fiscal decentralization and local government innovation preferences among provinces (including municipalities directly under the central government).

From a regional perspective, firms in the eastern region exhibit the highest overall innovation output, with a mean value of 121.772, approximately 1.8 times that of firms in the western region (67.597) and 1.6 times that of those in the central region (74.688). This gradient is fully consistent with the distribution of fiscal autonomy: eastern provinces have the highest level of fiscal decentralization, with a mean of 0.727, followed by the central region (0.486) and the western region (0.392).

From the perspective of ownership type, firm innovation performance shows significant ownership heterogeneity. The innovation intensity of SOEs is substantially higher than that of private enterprises. Consistent with the findings in Table 3, SOEs present not only a higher average innovation level but also systematically stronger innovation performance across the entire distribution. Based on Table 2, we observe that a small number of highly innovative firms dominate the overall statistics. Further analysis reveals that innovation among private enterprises is highly concentrated in a small number of high-quality firms, whereas SOEs display a more broadly distributed innovation capacity.

**Table 2. Descriptive statistics table.**

| Variables | Observed value | Mean | Standard deviation | Minimum value | Maximum value | Median |
|---|---|---|---|---|---|---|
| *Innovation* | 9 180 | 107.918 | 203.622 | 1 | 4502 | 82 |
| *Fisdec* | 9 180 | 0.649 | 0.187 | 0.251 | 0.926 | 0.714 |
| *Prefer* | 9 180 | 0.034 | 0.017 | 0.008 | 0.068 | 0.038 |
| *Herf* | 9 180 | 0.147 | 0.107 | 0.017 | 0.533 | 0.115 |
| *Cash* | 9 180 | 0.168 | 0.112 | 0.019 | 0.560 | 0.139 |
| *Debt* | 9 180 | 0.437 | 0.199 | 0.060 | 0.929 | 0.433 |
| *Age* | 9 180 | 2.336 | 0.593 | 0.768 | 3.273 | 2.381 |
| *Size* | 9 180 | 22.378 | 1.218 | 20.134 | 26.080 | 22.218 |
| *GDP* | 9 180 | 11.131 | 0.453 | 10.184 | 12.013 | 11.146 |
| *FDI* | 9 180 | 8.456 | 1.200 | 5.825 | 10.580 | 8.569 |
| *Finance* | 9 180 | 0.083 | 0.040 | 0.034 | 0.199 | 0.071 |
| *Industry* | 9 180 | 0.531 | 0.117 | 0.359 | 0.839 | 0.503 |
| *State* | 9 180 | 0.406 | 0.491 | 0 | 1 | 0 |

**Table 3. Descriptive statistics for key variables table.**

| Variables | Region | | | Nature of property rights | |
|---|---|---|---|---|---|
| | Eastern region | Central region | Western region | SOEs | Private firms |
| *Innovation*(mean) | 121.772 | 74.688 | 67.597 | 50.301 | 19.095 |
| *Innovation* (median) | 91 | 46 | 37 | 45 | 7 |
| *Fisdec* (mean) | 0.727 | 0.486 | 0.392 | 0.626 | 0.659 |
| *Fisdec* (median) | 0.626 | 0.659 | 0.626 | 0.659 | 0.626 |
| **Observed** | 6651 | 1388 | 1141 | 2819 | 6361 |

## 4. Empirical results and analysis

### 4.1. Benchmark regression results

Table 4 shows the results of the benchmark regression of the impact of fiscal decentralization on technological innovation of enterprises. Column (1) of Table 4 controls only for enterprise fixed effects and year fixed effects, while column (2) of Table 4 introduces control variables on the basis of column (1). From the benchmark regression results, it can be seen that the regression coefficient of fiscal decentralization is significantly negative at the 1% level, indicating that fiscal decentralization reduces fiscal decentralization reduces the number of enterprise patent applications and significantly inhibits the level of enterprise technological innovation. Hypothesis H1 is verified.

### 4.2. Robustness test

#### 4.2.1. Replacement of core explanatory variables.
Fiscal autonomy indicator can simultaneously reflect the inter-period changes and inter-regional differences in the central-local relationship, and is a relatively better measure of fiscal decentralization, and fiscal decentralization is measured by the degree of fiscal autonomy in the benchmark regression results. Based on the study by Jin et al., fiscal revenue decentralization captures the share of own revenues of the three levels of governments (central, state/province/region and local) as a proportion of general government revenue [51]. Fiscal expenditure decentralization measures the share of spending taking place at the sub-national level (using all resources

Table 4. Benchmark regression results.

| Variables | (1) | (2) |
|---|---|---|
| Fisdec | -4.775***(0.094) | −1.253***(0.099) |
| Herf | | -0.047 (0.099) |
| Cash | | -0.159***(0.057) |
| Size | | 0.087***(0.011) |
| Debt | | -0.073(*) (0.044) |
| Age | | 0.437***(0.027) |
| GDP | | 0.823***(0.058) |
| FDI | | -0.012 (0.017) |
| Finance | | -0.698 (0.566) |
| Industry | | -0.099 (0.247) |
| Firm fixed effects | Yes | Yes |
| Year fixed effects | Yes | Yes |
| Constant | 7.498***(0.061) | −6.640***(0.534) |
| $R^2$ | 0.239 | 0.239 0.509 |
| N | 9 180 | 9 180 |

Note: *,**and***denote significance at the 10%, 5% and 1% levels, respectively; robust standard errors are in parentheses.

available, except borrowing) relative to total expenditure of the general government. In the benchmark regression results, fiscal decentralization is measured by the degree of fiscal autonomy. Here, the fiscal autonomy indicator is replaced by the revenue decentralization indicator and the expenditure decentralization indicator respectively to test the robustness of the benchmark regression results. Specifically, revenue decentralization is measured as the ratio of local fiscal revenue to total national fiscal revenue, while expenditure decentralization is measured as the ratio of local fiscal expenditure to total national fiscal expenditure. The regression coefficients of fiscal revenue decentralization and fiscal expenditure decentralization in columns (1) and (2) of Table 5 are both significantly negative at the 1% level, which is consistent with the benchmark regression results.

**4.2.2. Other robustness tests.** To test whether the baseline regression results are susceptible to other potential biases and identification concerns, this paper conducts a series of robustness tests, with the regression results presented in Table 6.

First, considering that firm-level innovation activities may exhibit intertemporal serial correlation and that unobserved firm-specific heterogeneous characteristics may persist across years, this paper clusters standard errors at the firm level. This adjustment can control for the problem of within-firm dependence and ensure that statistical inference will not be biased due to underestimated standard errors, with the corresponding regression results shown in Table 6, column (1). Second, to address the potential issue that unobservable region-specific time-varying factors may affect both fiscal decentralization and firm innovation simultaneously, this paper further controls for region × year fixed effects. This model specification can absorb all time-varying region-specific heterogeneous shocks, including differentiated regional development trajectories, innovation policy cycles, and macroeconomic fluctuations, thereby allowing firms in different regions to respond heterogeneously to common time shocks. The estimation results after incorporating region × year fixed effects are presented in Table 6, column (2). Finally, given the promulgation of relevant policies in China in 2016, which marked a major institutional adjustment in central-local fiscal relations, the behaviors of firms and local governments in that year may have reflected more transitional responses rather than stable policy effects. To avoid potential interference caused by such transitional dynamics, this paper excludes the sample observations of 2016 and re-estimates the model, with the relevant results shown in Table 6, column (3).

**Table 5. Replacement of core explanatory variables.**

| Variables | (1) Fiscal revenue decentralization | (2) Fiscal expenditure decentralization |
|---|---|---|
| *Fisdec* | -0.287***(0.021) | −0.085***(0.007) |
| *Herf* | -0.074 (0.099) | −0.067 (0.099) |
| *Cash* | -0.159***(0.057) | −0.152(***)(0.057) |
| *Size* | 0.095***(0.011) | 0.097(***)(0.011) |
| *Debt* | -0.072(*) (0.044) | −0.074(*)(0.044) |
| *Age* | 0.481***(0.026) | 0.495***(0.026) |
| *GDP* | 0.884***(0.058) | 0.845***(0.058) |
| *FDI* | 0.002 (0.017) | 0.012 (0.017) |
| *Finance* | 3.133 ***(0.660) | 0.871 (0.594) |
| *Industry* | -0.166 (0.246) | 0.086 (0.248) |
| Firm Fixed Effects | Yes | Yes |
| Year fixed effects | Yes | Yes |
| *Constant* | -8.292***(0.522) | −7.882(***)(0.522) |
| $R^2$ | 0.510 | 0.510 |
| *N* | 9 180 | 9 180 |

Note: *,** and***denote significance at the 10%, 5% and 1% levels, respectively; robust standard errors are in parentheses.

Across all model specifications, the regression coefficients of fiscal decentralization remain significantly negative at the 1% statistical level, which is consistent with the results of the benchmark regression, indicating that the findings of the benchmark regression in this paper are robust.

**4.2.3. Endogeneity test.** Omitted variables, measurement error of fiscal decentralization, and possible reverse causality between fiscal decentralization and firms' technological innovation can affect the empirical results. To mitigate the above endogeneity problems, this paper uses the interaction term between the geographic distance (*Distance*) and the time trend (*Year*) from the capital city of each province to Beijing (the central government) as an instrumental variable for fiscal decentralization (Table 7). Fiscal decentralization is not only the result of top-down institutional design, but also due to the "spatial attenuation" of public goods supply capacity and the local government's advantage of being located at [1,12]. The geographical distance to Beijing mainly reflects attributes of administrative and political distance: The geographical distance to Beijing mainly reflects attributes of administrative and political distance: the farther the local government is from Beijing, the higher the degree of fiscal decentralization it obtains, which satisfies the relevance requirement of instrumental variables. Meanwhile, as a natural factor, geographical distance also meets the exogeneity requirement of instrumental variables. This is because Beijing is not China's core commercial or technological innovation hub, and major domestic innovation centers are geographically distant from Beijing, meaning this distance does not directly affect firms' technological innovation. However, using only the indicator of geographical distance between each provincial capital and Beijing would result in the loss of time-varying information. Therefore, this paper introduces a time trend term. Time-invariant regional development characteristics are absorbed by regional fixed effects, while time-varying regional shocks are controlled by the interaction term of geographical distance between each provincial capital and Beijing × year fixed effects. This interaction term is then employed as the final instrumental variable.

**Table 6. Other robustness test results.**

| Variables | (1) | (2) | (3) |
|---|---|---|---|
| *Fisdec* | -1.253***(0.201) | −16.110***(2.685) | −1.278***(0.107) |
| *Herf* | -0.047 (0.223) | −0.080 (0.218) | −0.024 (0.108) |
| *Cash* | -0.159 (0.099) | −0.193(**) (0.098) | −0.173(***) (0.063) |
| *Size* | 0.087***(0.029) | 0.0837***(0.027) | 0.0845***(0.013) |
| *Debt* | -0.073 (0.083) | −0.005 (0.083) | −0.072 (0.048) |
| *Age* | 0.437***(0.059) | 0.151**(0.072) | 0.417***(0.029) |
| *GDP* | 0.823***(0.107) | 4.285 (3.233) | 0.852***(0.063) |
| *FDI* | -0.012 (0.030) | 2.227***(0.783) | 0.005 (0.019) |
| *Finance* | -0.698 (1.156) | 86.45 ***(21.850) | −0.745 (0.603) |
| *Industry* | -0.0985 (0.573) | −51.75***(17.570) | −0.270 (0.267) |
| *Constant* | -6.640***(1.089) | −33.260 (41.050) | −6.906***(0.581) |
| $R^2$ | 0.509 | 0.573 | 0.519 |
| *N* | 9 180 | 9 180 | 8 160 |

Note: *,**and***denote significance at the 10%, 5% and 1% levels, respectively; robust standard errors for clustering are in parentheses in column (1), and robust standard errors in parentheses in column (2) and column (3).

**Table 7. Endogeneity test results.**

| Variable | 2SLS |
|---|---|
| *Distance∗Year* | -4.163 ***(0.238) |
| *Herf* | -0.046 (0.098) |
| *Cash* | -0.282 ***(0.057) |
| *Size* | 0.069***(0.011) |
| *Debt* | -0.001 (0.044) |
| *Age* | -0.120***(0.036) |
| *GDP* | 0.780***(0.058) |
| *FDI* | -0.034**(0.017) |
| *Finance* | 1.438**(0.583) |
| *Industry* | 0.288 (0.246) |
| Firm Fixed Effects | Yes |
| Year fixed effects | Yes |
| *Constant* | -6.463 ***(0.610) |
| $R^2$ | 0.944 |
| *N* | 9 180 |
| F-statistic for the first period | 1 661.820*** |

Note: *,**and***denote significance at the 10%, 5% and 1% levels, respectively; robust standard errors are in parentheses.

## 5. Mechanism test and analysis

According to the theoretical analysis in the previous paper, the innovation preference of local governments has a mediating role between fiscal decentralization and technological innovation of enterprises, and there is a property right

dependence characteristic. This paper refers to the two-step regression method proposed by Zhao et al. to set up a regulated mediation model to empirically test the relationship between fiscal decentralization, local government innovation preference, enterprise technological innovation and property right nature [23]. In the first step, the impact of fiscal decentralization (*Fisdec*) on local government innovation preference (*Prefer*) is tested; in the second step, tests the impact of local government innovation preference (*Prefer*) on enterprise technological innovation (*nnovation*) and the moderation of the relationship between the nature of property rights. Based on the above ideas, the study sets up the following models in turn:

$$Prefer_{i,t} = \beta_0 + \beta_1 Fisdec_{i,t} + \beta_2 GDP_{i,t} + \beta_3 FDI_{i,t} + \beta_4 Finance_{i,t} + \beta_5 Industry_{i,t} \tag{2}$$

$$Innovation_{i,t} = \gamma_0 + \gamma_1 Fisdec_{i,t} + \gamma_2 Prefer_{i,t} + \gamma_3 State_{i,t} + \gamma_4 State_{i,t} \times Prefer_{i,t} + \gamma_5 Herf_{i,t} + \gamma_6 Cash_{i,t} + \gamma_7 Size_{i,t} + \gamma_8 Debt_{i,t}$$
$$+ \gamma_9 Age_{i,t} + \gamma_{10} GDP_{i,t} + \gamma_{11} FDI_{i,t} + \gamma_{12} Finance_{i,t} + \gamma_{13} Industry_{i,t} \tag{3}$$

All variables in model (2) are regional level (provincial) variables, and i stands for province. In model (2) and model (3), QUOTE *Prefer Prefer* denotes local government innovation preference, QUOTE *State State* denotes the nature of ownership of the actual controller, and the rest of the variables are the same as in model (1). In order to exclude the interference of extreme values on the regression results, this paper shrinks the upper and lower 1% of all continuous variables.

The results of the mechanism test are shown in Table 8. The regression results show that the effects of fiscal decentralization on local government innovation preference ($\beta_1$= 0.034, *P*< 1%) and the effects of the cross-multiplier term of local government innovation preference and the nature of property rights on enterprise technological innovation ($\gamma_4$= 6.676, *P*< 1%) pass the significance test at the 1% level, indicating that the moderated mediation effect holds. The moderated mediation effect of local government innovation preference =$\beta_1(\gamma_2+\gamma_4 State)$ = 0.034 $\times$ (–2.941 + 6.676State). When the value of property right nature is 0 (i.e., the property right nature is private), the mediation effect of local government innovation preference is −0.010, which indicates that there is a negative mediation effect of local government innovation preference between fiscal decentralization and technological innovation of private enterprises. When the value of property right nature is 1 (i.e., the property right nature is state-owned), the mediation effect of local government innovation preference is 0.127, indicating that there is a positive mediation effect of local government innovation preference between fiscal decentralization and technological innovation of state-owned enterprises. From the differences in the mediation effect of local government innovation preference under different property rights, it can be found that the mediation effect of local government innovation preference is regulated by the nature of property rights, and there is a significant property rights-dependent feature. Fiscal decentralization promotes the technological innovation of state-owned enterprises and inhibits the technological innovation of private enterprises by influencing the innovation preference of local governments, which proves the effectiveness of state-owned enterprises as the core institutional arrangement of the socialist market economic system, and hypothesis H3 is verified. The mediating effect of local government innovation preference between fiscal decentralization and technological innovation of enterprises under both state-owned and private property rights holds, and hypothesis H2 is verified.

## 6. Conclusion and policy implications

### 6.1. Research conclusion

Based on the panel data of A-share listed companies that regularly publish annual financial reports during the period of 2015–2023, this paper empirically examines the mechanism of Chinese-style fiscal decentralization on technological innovation of enterprises and the heterogeneity of property rights by using a moderated mediation model. The study draws the following conclusions: first, overall, fiscal decentralization has a significant inhibitory effect on corporate technological

**Table 8. Mechanism test results.**

| Variables | First step test: model (1) | Second step test: model (2) |
|---|---|---|
| | Dependent Variable: *Prefer* | Dependent variable: *Innovation* |
| *Fisdec* | 0.034***(0.004) | −1.250***(0.100) |
| *Prefer* | | -2.941***(0.828) |
| *State* | | -0.167***(0.056) |
| *Prefer*State* | | 6.676***(1.115) |
| *Herf* | | -0.078 (0.099) |
| *Cash* | | -0.157***(0.057) |
| *Size* | | 0.088***(0.011) |
| *Debt* | | -0.061 (0.044) |
| *Age* | | 0.451***(0.028) |
| *GDP* | 0.018***(0.002) | 0.804***(0.060) |
| *FDI* | -0.001 (0.001) | −0.004 (0.017) |
| *Finance* | -0.073*(0.039) | −0.867 (0.587) |
| *Industry* | 0.019 (0.013) | −0.085 (0.248) |
| Province fixed effects | Yes | Yes |
| Year fixed effects | Yes | Yes |
| *Constant* | -0.181***(0.020) | −6.462***(0.567) |
| $R^2$ | 0.531 | 0.531 |
| *F* | 9.610*** | 128.100*** |
| *N* | 279 | 9 180 |

Note: *,**and***denote significance at the 10%, 5% and 1% levels, respectively; robust standard errors are in parentheses.

innovation, which still holds after a series of robustness tests and endogeneity tests. Second, local government innovation preference has a mediating role between fiscal decentralization and enterprise technological innovation, and fiscal decentralization indirectly affects enterprise technological innovation by influencing local government innovation preference. This is consistent with the institutional condition-dependent findings from the cross-country study by Carrillo-Pulgar et al., who show that decentralization is positively associated with environmental spending, provided that institutional quality is high [52]. This study finds that decentralization may lead to short-term behavior in environments with low institutional quality, whereas in environments with high institutional quality, decentralization can improve efficiency through informational advantages. Third, the mediating effect of local government innovation preference is characterized by significant property rights dependence, and fiscal decentralization promotes technological innovation of state-owned enterprises and inhibits technological innovation of private enterprises through influencing local government innovation preference, which proves the effectiveness of state-owned enterprises as the core institutional arrangement of the socialist market economic system.

## 6.2. Policy implications

In his article "Adherence to Historical Materialism Continuously Opens New Frontiers for the Development of Marxism in Contemporary China", General Secretary Xi Jinping, for the first time, juxtaposed the roles and counterproductive effects between the productive forces and the relations of production, emphasizing that "the productivity criterion must

be understood comprehensively and accurately, and cannot be absolutized, and the productivity criterion must not be understood by setting aside the relations of production and superstructure". Fiscal decentralization belongs to the scope of the relations of production, how to realize the synergy of interests and convergence of objectives of all the main parties in the Chinese fiscal decentralization and its embedded "central government-local government-enterprise" three-layer commission-agent relationship, is the key to promote technological innovation of enterprises to form a new quality of productivity. This is the key to promoting technological innovation and forming new quality productivity. This pattern is consistent with the political competition mechanism documented by Yu et al., in which Chinese local governments engage in various strategic behaviors to improve their standing in relative performance evaluation [14]. The nonlinear relationship between decentralization and innovation identified in this study helps local governments implement policy initiatives preferred by the central government, thereby signaling both policy compliance and governance capacity, and consolidating their advantageous position in the promotion tournament.Furthermore, the impact of fiscal decentralization on innovation exhibits property right dependence. For policymakers in other centralized systems, granting local fiscal autonomy without aligning local incentives with innovation goals may reproduce the distortion of "prioritizing production over innovation" that once occurred in China. Conversely, for federal countries seeking to strengthen central coordination, China's experience demonstrates that performance based promotion tournaments can be effective, but also indicates that such mechanisms may induce short-term behavior if performance criteria are poorly designed.

First, promote the reform of the central-local relationship and mobilize the "two positives" of the central and local governments. First, give full play to the overall coordinating role of the new national system, highlighting the leading role of the central government in basic research and major scientific and technological facilities, and effectively organizing and integrating the behavior of local governments to achieve the goal of national development of new quality productive forces. Second, to establish a sound financial system that matches the rights and responsibilities of the central government with those of the local government, to change the situation of serious asymmetry between the rights and responsibilities of the local government caused by the upward transfer of financial rights and the downward sinking of the rights after the reform of the tax-sharing system, and to shift from large-scale transfers to the granting of its own revenues that match the expenditures of the local government. Finally, there is a certain degree of alternation between economic growth and technological innovation, and the relative incentive intensity of economic growth tasks should be further reduced in the context of accelerating the development of new-quality productive forces to prevent major incentive deviations.

Second, build a new type of government-enterprise relationship, and increase the support for technological innovation of enterprises. First of all, accelerate the construction of a new type of government-enterprise relationship that is "pro" and moderate, "clear" and active, and while stimulating the enthusiasm and initiative of enterprises in technological innovation, the government will not set up rents and enterprises will not seek rents, so as to avoid the policy arbitrage that may be triggered in the process of the development of new quality productivity. Secondly, technological innovation has a very strong and positive impact on the development of the new quality productivity. Secondly, technological innovation has a strong public goods attribute, which leads to a lack of effective technological innovation supply for private enterprises that rely excessively on market mechanisms, and private enterprises can be guided to carry out the national innovation will and goals through the embedding of party organizations in the corporate governance structure. Finally, specialized, special and new small and medium-sized enterprises, with their advantages in specialization and technological innovation, are a key force in the development of new productive forces, and can be nurtured and supported through a combination of policies such as tax incentives, financial subsidies, and market access, as well as through the optimization of the business environment.

Thirdly, clarify the function and position of state-owned enterprises, and play the role of a model for technological innovation. First of all, SOEs have the dual attributes of profit-making and fulfilling the national mission, which is an important institutional arrangement to overcome the "market failure" in technological innovation, and should play an advantageous role in the organizational form in the national development strategy and the new national system prompted by the new

quality of productive forces. Secondly, SOEs have non-market contractual characteristics, and need to implement residual claim incentives for managers and employees to improve income expectations, strengthen self-supervision and mutual supervision, and increase the willingness to take risks, so as to promote the technological innovation capacity of SOEs. In terms of the implementation of residual claim incentive programs, practices such as employee stock ownership plans, excess profit sharing mechanisms, virtual restricted equity and time unit plans (TUPs) can be applied to industries and fields controlled by the state-owned economy. Finally, SOEs should be guided to increase their economic deployment in strategic emerging industries such as biomanufacturing, commercial spaceflight, low-space economy, and future industries such as quantum and life sciences to create "chain-length" SOEs, and to make good examples of "neck-breaking" technologies, industry-common technologies, original technologies and disruptive technological innovations. It will be a model of "neck-breaking" technology, industry common technology, original technology and subversive technology innovation, and promote the development of new quality productive forces by cooperating with private enterprises in a way of bringing small and large enterprises together.

## Supporting information

**S1 File. Minimal data set.**
(XLSX)

## Author contributions

**Conceptualization:** Jie Tu.

**Data curation:** Jie Tu.

**Formal analysis:** Jie Tu.

**Investigation:** Jie Tu.

**Methodology:** Jie Tu.

**Software:** Jie Tu.

**Validation:** Jie Tu.

**Visualization:** Jie Tu.

**Writing – original draft:** Jie Tu.

**Writing – review & editing:** Jie Tu.

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
