## [Decision Letter · Decision Letter 0]

11 Dec 2025

Dear Dr. Tu,

Thank you for submitting your manuscript to PLOS ONE. After careful consideration, we feel that it has merit but does not fully meet PLOS ONE’s publication criteria as it currently stands. Therefore, we invite you to submit a revised version of the manuscript that addresses the points raised during the review process.

We look forward to receiving your revised manuscript.

Kind regards,

Tatchalerm Sudhipongpracha

Academic Editor

PLOS One

Additional Editor Comments (if provided):

Reviewers' comments:

Reviewer's Responses to Questions

**Comments to the Author**

1. Is the manuscript technically sound, and do the data support the conclusions?

Reviewer #1: Partly

Reviewer #2: Yes

2. Has the statistical analysis been performed appropriately and rigorously?

Reviewer #1: Yes

Reviewer #2: Yes

3. Have the authors made all data underlying the findings in their manuscript fully available?

Reviewer #1: No

Reviewer #2: Yes

4. Is the manuscript presented in an intelligible fashion and written in standard English?

Reviewer #1: Yes

Reviewer #2: Yes

Reviewer #1: The reviewed article examines an intriguing topic: the role of local governments in supporting innovation, particularly in enterprises they own. However, this issue is specific to China for two main reasons. First, local governments own state-owned companies (SOCs) that operate in private, and even international, markets—not just in areas related to municipal services. Second, the Chinese model of decentralization is distinct; it constitutes a form of delegation rather than true decentralization as defined by fiscal decentralization theory. While the authors touch on this in the article, it should have been emphasized in the title. Additionally, the article should explain why this topic might interest readers from other countries where such solutions do not exist.

One issue is that the authors claim to focus on technology companies, but this is not evident in the sample presented. They refer to patents as a measure of technological innovation; however, patents can address a wide range of issues beyond just innovative technological solutions. It might be more appropriate to discuss innovative companies in general, rather than strictly focusing on innovations in new technologies. Alternatively, it could be helpful to clarify that this issue pertains specifically to new technologies.

Another concern is the lack of a reference that highlights a characteristic feature of the Chinese model of power division—namely, the competition among local authorities for the competition among local authorities for "points" from the central government. (see for example Yu, C., Hou, L., Lyu, Y., & Zhang, Q. (2022). Political competition, spatial interactions, and default risk of local government debts in China. Papers in Regional Science, 101(3), 717–743. Retrieved from https://doi.org/10.1111/pirs.12668) I believe this is a crucial factor in understanding why local authorities support actions favored by the central government, as evidenced by the quote at the line 522.

Here are a few additional suggestions:

1. Including a table that lists the names and descriptions of the variables would enhance navigation throughout the article.

2. The statistics table should present the data before logarithmization, as it is important and interesting to see for example the actual number of patents filed by individual companies and the extent of variation. If I calculated correctly, the maximum is about 800 patents per year, which is a significant figure. It may be worth exploring what type of company is responsible for such a remarkable level of innovation.

Reviewer #2: This manuscript explores the effects of fiscal decentralization on enterprise-level technological innovation in China through a moderated mediation model embedded within a three-tier principal-agent framework involving central government, local governments, and enterprises. The topic is timely and relevant for local governance and fiscal policy studies. The empirical strategy is clearly articulated and supported by robustness checks and instrumental variable analysis. However, there are several areas that should be improved. The following are my detailed comments.

My first concern is that while the manuscript invokes a three-tier “principal-agent” framework, the conceptual clarity is lacking. Theoretical underpinnings (i.e., linkage between fiscal decentralization and innovation suppression) are insufficiently grounded in local governance literature. The authors mostly cite economic literature, overlooking well-established debates in public administration and fiscal federalism regarding vertical accountability, local autonomy, and institutional capacity. The author may enhance the theoretical framework by integrating key works in local governance and decentralization theory. Please also consider how institutional quality, local political capacity, and discretion mediate the impact of decentralization on policy outcomes.

Second, the use of “local government innovation preference” as a mediating variable is both central and problematic. Its operationalization via the ratio of science & tech expenditure to general budget lacks depth and does not convincingly reflect "preference"—a psychological and political orientation—especially when interpreted as a mediating institutional mechanism.

Third, the author employs the interaction between geographic distance to Beijing and a time trend as an instrument for fiscal decentralization. While this aligns with some spatial attenuation logic, the justification for exclusion restriction remains weak. Geographic distance could plausibly correlate with regional developmental trajectories, market access, and innovation potential. This undermine the exogeneity assumption.

Fouth, the manuscript idealizes SOEs as effective agents of innovation without sufficient critical reflection. It claims SOEs overcome “market failure” via dual coordination mechanisms but downplays inefficiencies, rent seeking, or governance challenges associated with SOEs in China and elsewhere. Please balance the discussion with references to literature on SOE inefficiencies and performance heterogeneity.

.

Reviewer #1: No

Reviewer #2: No

---

## [Author Response · Author response to Decision Letter 1]

7 Jan 2026

We are deeply honored and grateful to have the opportunity to improve the quality of our manuscript. We sincerely appreciate your constructive comments, insightful suggestions, and words of encouragement and recognition.

We have carefully studied and fully incorporated all the review comments, and accordingly conducted a comprehensive revision of the manuscript—all the revised content is included in the appendix. Thank you for your valuable and profound insights, which have effectively helped us grasp the methods and logic for writing high-quality academic papers and facilitated the continuous refinement of our work.

Hereby, we extend our most sincere gratitude to you!

---

## [Decision Letter · Decision Letter 1]

11 Feb 2026

Dear Dr. Tu,

Thank you for submitting your manuscript to PLOS ONE. After careful consideration, we feel that it has merit but does not fully meet PLOS ONE’s publication criteria as it currently stands. Therefore, we invite you to submit a revised version of the manuscript that addresses the points raised during the review process.

We look forward to receiving your revised manuscript.

Kind regards,

Tatchalerm Sudhipongpracha

Academic Editor

PLOS One

Journal Requirements:

Reviewers' comments:

Reviewer's Responses to Questions

**Comments to the Author**

Reviewer #1: All comments have been addressed

Reviewer #2: All comments have been addressed

2. Is the manuscript technically sound, and do the data support the conclusions?

Reviewer #1: Partly

Reviewer #2: Yes

3. Has the statistical analysis been performed appropriately and rigorously?

Reviewer #1: No

Reviewer #2: Yes

4. Have the authors made all data underlying the findings in their manuscript fully available?

Reviewer #1: Yes

Reviewer #2: Yes

5. Is the manuscript presented in an intelligible fashion and written in standard English?

Reviewer #1: Yes

Reviewer #2: Yes

Reviewer #1: The authors addressed most of the comments, but I still have some doubts:

1. For example, I wonder if the logic presented in the introduction about the negative impact of decentralization on innovation under the Chinese model of decentralization is valid. (lines 85-89) After all, if we have a system in which local authorities strive to obtain "points" from the state, they will decide to implement innovative activities as long as they are consistent with state policy. Moreover, they will decide to implement such activities even against the needs of residents – because it is not their needs, but the center's recommendations that are being implemented in such circumstances. Moreover, it is precisely this logic that the authors demonstrate in their conclusion.

2. In the previous comments, I noted that, given the specificity of China, one should ask why its analysis might be interesting to an international audience. The only argument the authors managed to find referred to readers from other centralized countries. This is not convincing. In my opinion, it would be worthwhile, for example, to demonstrate in the conclusions whether and how these conclusions can/cannot be transferred to other countries with different decentralization systems (what other forms of support for specific actions important for the entire country can state authorities undertake in other countries, and whether this is effective – I recommend, for example, literature on decentralization and environmental protection).

3. Thank you for organizing the variables in the table. This makes it much easier to follow the models, and it clearly shows that the descriptions of variables that are crucial to the work's purpose are missing definitions. Specifically, how is the fiscal autonomy indicator measured – where is its definition? Similarly, I don't see a definition of the revenue decentralization indicator or the expenditure decentralization indicator.

4. The authors briefly comment on the variables' statistics, but they do so in a very mechanical manner. From the perspective of the article's purpose, it would be interesting to show and discuss the differences and correlations of the most important variables, combined with detailed information. For example, does any particular region stand out in the statistics (in terms of the number of patents), or do the statistics show differences between patents filed by SOEs and private companies. Which companies (by name) are among those filing the most patents? Perhaps some industries stand out?

5. With regard to models and inferences about the influence of local governments on SOEs vs. private companies, it would be reasonable to conduct separate models for these types of companies and also, in the initial models (before adding the mediation mechanism), include the interaction of ownership type with the level of decentralization

Reviewer #2: Thank you for addressing all my previous comments. I really appreciate your efforts. Therefore, I recommend accepting this paper in its current version.

.

Reviewer #1: No

Reviewer #2: No

---

## [Author Response · Author response to Decision Letter 2]

3 Mar 2026

Dear Editor and Reviewers,

We are greatly honored and sincerely grateful for the opportunity to revise and improve our manuscript. We highly appreciate your constructive comments, insightful suggestions, and encouraging remarks. We have carefully reviewed all comments and revised the manuscript accordingly. The detailed revisions are listed below or attached in the file Responses to Reviewers.

Thank you for your valuable and profound comments, which have effectively helped us learn how to write high-quality academic papers and enhance our manuscript. We would like to express our sincere gratitude again.

Response to valuable comment 1

We greatly appreciate your comment that our discussion of the negative impact of China’s fiscal decentralization on innovation remains insufficient. Accordingly, we have supplemented relevant arguments on how decentralization inhibits innovation in the manuscript (Please see P.3 and P.5).

Specifically, under China’s decentralization model, it is true that—as you pointed out—local authorities will prioritize innovation activities as long as they meet the criteria for officials to gain performance points. However, innovation activities are characterized by large investment scale, high risk, and vulnerability to imitation and infringement. The scientific and technological innovation output in one region can promote the economic and social development of other regions in a short period. Such positive externalities often lead local governments to lack incentives for innovation and tend to reduce government spending on science and technology.

Under the current performance evaluation system, local governments are constrained by limited time, effort, and resources, making it difficult to simultaneously achieve multiple goals such as economic growth and innovation-driven development. This results in selective implementation and goal substitution, where local officials prioritize short‑term visible GDP growth over long‑term investment in innovation.

Response to valuable comment 2

We greatly appreciate your valuable suggestion that the manuscript can be further improved in elaborating on the international relevance of studying the impact of fiscal decentralization on innovation in the Chinese context. In response, we have enriched the relevant discussion in the manuscript (Please see P.4 and P18).

Specifically, the “decentralization-inhibiting-innovation” effect revealed in this study is not unique to China, but a general outcome of the mismatch between fiscal decentralization and political incentive structures. Your advice has guided our revision. After reviewing the related literature, we find that in the field of environmental protection, cross-country studies have confirmed that decentralization may lead to a “race to the bottom”. This study extends this logic to innovation policy, showing that when local governments face competing multiple tasks and innovation is not yet included in core performance assessments, decentralization may induce short-term behavior.

This finding is informative for developing countries undertaking decentralization reforms (e.g., Indonesia, India) and federal countries seeking to optimize central-local relations (e.g., the United States, Germany). The design of decentralization needs to be aligned with central policy orientations and performance evaluation systems, rather than simply pursuing the maximization of decentralization.

In addition, China’s experience shows that selective decentralization under political centralization can stimulate innovation through central policy adjustments (e.g., the innovation-driven development strategy), thus providing complementary evidence from a non-Western context.

Response to valuable comment 3

Thank you for your valuable suggestion regarding the clarification of variable definitions. We fully agree that clear operationalization of key variables is essential for the replicability and transparency of our empirical analysis.

In response to your comment, we have made the following revisions to the manuscript:

We have supplemented the detailed definition and measurement approach of fiscal autonomy in 3.1.2 Explanatory Variables (Please see P.10). This indicator aligns with the standard approach in the fiscal federalism literature and effectively captures the extent to which local governments possess independent fiscal decision-making power. Furthermore, we have updated TABLE 1 Variables and Definitions to ensure that readers can clearly understand how this core explanatory variable is constructed.

Since the two indicators of revenue decentralization and expenditure decentralization are mainly used as alternative proxy variables for fiscal autonomy to test the robustness of the baseline regression results, rather than as core explanatory variables in the main analysis, we supplement their detailed definitions and measurement methods in 4.2.1 Replacement of Core Explanatory Variable (Please see P.14) .

Thank you again for your constructive feedback, which has helped us improve the clarity and rigor of our empirical framework.

Response to valuable comment 4

Thank you for your valuable suggestions on the descriptive statistics. In response to your comments, we have substantially revised Section 3.4 Descriptive Statistics of Main Variables (Please see P.13).

The newly added stylized facts of key variables in the manuscript, especially the co-movement between regional fiscal autonomy and innovation capacity, as well as the significant divergence in innovation intensity between state-owned enterprises and private enterprises under similar fiscal environments—provide direct support for our hypothesis regarding the conditional effects of fiscal decentralization on firm innovation.

We believe these revisions have addressed your concerns and significantly strengthened the empirical foundation of this paper.

Response to valuable comment 5

I greatly appreciate your suggestion to conduct separate regression estimations for SOEs and private enterprises. The moderated mediation model used in this study can directly test the boundary conditions of the mediating mechanism. Rather than only showing that the effect of fiscal decentralization differs between SOEs and private enterprises, it explicitly examines “why and how ownership matters”, through the heterogeneous response to local government innovation preferences.

The interaction term between local government innovation preference and ownership type (Prefer×State, coefficient=6.676***) is statistically significant, indicating that the mediating mechanism varies across ownership types. This provides deeper theoretical insights than separate subsample baseline models.

Second, the full-sample model in this study uses a total of 9180 observations, allowing for precise estimation of firm fixed effects and year fixed effects. Splitting the sample would reduce statistical power. The current model maximizes the use of available data while still identifying ownership heterogeneity effects.

Third, our theoretical framework emphasizes that SOEs and private enterprises face different incentives and constraints within the same institutional environment. The moderated mediation model allows the effect of the core channel (local government innovation preference) to vary with ownership type while controlling for the overall effect of fiscal decentralization. This is consistent with the theoretical framework: ownership type moderates the translation of government preferences into innovation outputs, rather than fundamentally altering the relationship between fiscal decentralization and innovation.

In summary, the current approach not only identifies whether heterogeneous effects exist, but also clarifies the channels and directions of such differences, which is crucial for policy design under China’s unique institutional context.

Response to valuable comment 6

We would like to express our sincere gratitude for recognizing our revision efforts and recommending acceptance of our manuscript in its current version. We deeply appreciate the time and dedication you devoted to providing thorough and constructive feedback throughout the review process. Your insightful comments have been invaluable in helping us strengthen our work and improve the overall quality of this paper. Thank you for your scholarly guidance and professional support.

---

## [Decision Letter · Decision Letter 2]

27 Mar 2026

Fiscal Delegation, Local Government Innovation Preference, and Enterprise Technological Innovation: Evidence from China

PONE-D-25-42189R2

Dear Dr. Tu,

We’re pleased to inform you that your manuscript has been judged scientifically suitable for publication and will be formally accepted for publication once it meets all outstanding technical requirements.

Kind regards,

Tatchalerm Sudhipongpracha

Academic Editor

PLOS One

Additional Editor Comments (optional):

Reviewers' comments:

Reviewer's Responses to Questions

**Comments to the Author**

Reviewer #1: All comments have been addressed

2. Is the manuscript technically sound, and do the data support the conclusions?

Reviewer #1: Yes

3. Has the statistical analysis been performed appropriately and rigorously?

Reviewer #1: Yes

4. Have the authors made all data underlying the findings in their manuscript fully available?

Reviewer #1: Yes

5. Is the manuscript presented in an intelligible fashion and written in standard English?

Reviewer #1: Yes

Reviewer #1: Thank you for considering my previous comments. I have no additional comments to add.

One aspect to consider for future studies is the measurement of innovation, particularly regarding long-term activities, through patent counts. Existing literature suggests that patents do not always equate to actual innovation, and not all innovations are protected by patents.

.

Reviewer #1: No

---

## [Editor Report · Acceptance letter]

PONE-D-25-42189R2

PLOS One

Dear Dr. Tu,

I'm pleased to inform you that your manuscript has been deemed suitable for publication in PLOS One. Congratulations! Your manuscript is now being handed over to our production team.

Kind regards,

on behalf of

Dr. Tatchalerm Sudhipongpracha

Academic Editor

PLOS One